# Indexes of Angiogenic Activation in Myocardial Samples of Patients with Advanced Chronic Heart Failure

**DOI:** 10.3390/medicina55120766

**Published:** 2019-11-29

**Authors:** Klara Komici, Isabella Gnemmi, Claudia Sangiorgi, Fabio Luigi Massimo Ricciardolo, Mauro Rinaldi, Antonino Di Stefano, Ermanno Eleuteri

**Affiliations:** 1Department of Medicine and Health Sciences, University of Molise, 86100 Campobasso, Italy; 2Pulmonary Rehabilitation Unit and Laboratory of Cytoimmunopathology of the Heart and Lung, Istituti Clinici Scientifici Maugeri, 28010 Veruno, Italy; isabella.gnemmi@icsmaugeri.it (I.G.); san.85@hotmail.it (C.S.); antonino.distefano@icsmaugeri.it (A.D.S.); 3Department of Clinical and Biological Sciences, University of Torino, San Luigi Hospital, 10043 Turin, Italy; fabiolugimassimo.ricciardolo@unito.it; 4Department of Cardiovascular and Thoracic Surgery, University of Turin, 10126 Turin, Italy; mauro.rinaldi@unito.it; 5Division of Cardiology, Istituti Clinici Scientifici Maugeri, 28010 Veruno, Italy; ermanno.eleuteri@icsmaugeri.it

**Keywords:** heart failure, angiogenesis, angiopoietin-1, angiopoietin-2, cardiac fibrosis

## Abstract

*Background and objectives*: Ischemic and idiopathic heart failure are characterized by reactive cardiac fibrosis and impaired vasculogenesis involving pro-angiogenic factors such as angiogenin, angiopoietin-1 (Ang-1), and angiopoietin-2 (Ang-2), as demonstrated in experimental models of heart failure. However, differences in the molecular pathways between these cardiomyopathies are still unclear. In this short communication, we evaluate and compare the expression of pro-angiogenic molecules in the heart tissue of patients with advanced chronic heart failure (CHF) of ischemic vs. nonischemic etiology. *Materials and Methods*: We obtained heart tissue at transplantation from left ventricular walls of 16 explanted native hearts affected by either ischemic (ICM) or nonischemic dilated cardiomyopathy (NIDCM). Tissue samples were examined using immunohistochemistry for angiogenic molecules. *Results*: We found immunopositivity (I-pos) for angiopoietin-1 mainly in the cardiomyocytes, while we observed I-pos for Ang-2 and Tie-2 receptor mainly in endothelial cells. Expression of Procollagen-I (PICP), angiogenin, Ang-1, and Tie-2 receptor was similar in ICM and NIDCM. In contrast, endothelial immunopositivity for Ang-2 was higher in ICM samples than NIDCM (*p* = 0.03). *Conclusions*: In our series of CHF heart samples, distribution of Ang-1 and angiogenin was higher in cardiomyocytes while that of Ang-2 was higher in endothelial cells; moreover, Ang-2 expression was higher in ICS than NIDCM. Despite the small series examined, these findings suggest different patterns of angiogenic stimulation in ICM and NIDCM, or at least a more altered endothelial integrity in ICD. Our data may contribute to a better understanding of the angiogenesis signaling pathways in CHF. Further studies should investigate differences in the biochemical processes leading to heart failure.

## 1. Introduction

Ischemic and nonischemic heart failure (HF) are characterized by reactive cardiac fibrosis and impaired vasculogenesis [1]. Angiogenesis is an essential process involved in the pathophysiological responses to a number of injuries such as gastrointestinal ulcer, stroke, myocardial infarction, and left ventricular hypertrophy [2,3,4]. It is regulated by Vascular Endothelial Growth Factor (VEGF) and by the angiopoietins-Tie (Ang-Tie) signaling pathway [5,6]. Angiopoietin-1 (Ang-1) and angiopoietin-2 (Ang-2) are vascular growth factors expressed mainly on endothelial cells. Ang-1 induces Tie-2 receptor activation and facilitates endothelial cell sprouting and vascular network maturation. Ang-2 also binds to Tie-2 receptor but inhibits Ang-1-Tie signaling by blocking Ang-1-induced phosphorylation of Tie-2, resulting in vascular destabilization and remodeling [7,8]. Angiogenin, a member of the ribonuclease (RNAse) superfamily, is a potent inducer of neovascularization in vivo, and its circulating levels reflect various angiogenic activities that include increased vessel permeability, endothelial proliferation and vascular maturation [9]. While idiopathic HF may have different triggers (e.g., immunological, metabolic, or genetic) the current pathophysiological model of ischemic HF is focused on the atherosclerosis process and impairment of angiogenesis. Data from experimental models and evaluation of circulating levels of angiogenesis factors in patients with idiopathic dilative cardiomyopathy report abnormal angiogenesis associated to cardiac remodeling and HF progression [10,11]. On the other hand, studies on the role of Ang-2 in the atherosclerosis process have reported conflicting results: Ang-2 antibodies administration leads to inhibition of atherosclerotic plaque progression, and Ang-2 seems to have a protective role against LDL oxidation [12,13]. Most of the mentioned studies, however, were performed on experimental models, and data that describe the vasculogenesis pattern in human heart tissue of end-stage HF are lacking. Furthermore, the differences in molecular pathways between idiopathic and ischemic cardiomyopathy are still unclear. Therefore, in this study we aimed to evaluate and compare the expression of proangiogenic molecules in the heart tissue of patients with advanced CHF of ischemic vs. idiopathic etiology.

## 2. Materials and Methods

This study was carried out in accordance with the recommendations of the ethical committee of the Fondazione Salvatore Maugeri, IRCCS, Pavia, Italy. The protocol was previously approved on 05.02.2007 by the Central Ethics Committee (CEC) of the Fondazione Salvatore Maugeri, Pavia, Italy (n.382 CEC). All subjects gave written informed consent for the study, which was carried out in accordance with the principles of the Declaration of Helsinki.

### 2.1. Heart Tissue Specimens from Ischemic and non-Ischemic Dilated Cardiomyopathy Patients Study Population

Nine male patients affected by ischemic dilated cardiomyopathy (ICM) and seven patients with nonischemic dilated cardiomyopathy (NIDCM) were included in our study. The mean age for ICM and NIDCM groups was, respectively, 60.8 ± 1.9 and 59.6 ± 2.4 years. ICM patients were all males, while the NIDCM group consisted of four males and three females. All patients presented an indication for heart transplantation according to the European Society of Cardiology Guidelines for the treatment of CHF, i.e., they all had end-stage HF with left ventricular ejection fraction <30%, and were in NYHA class III or IV at rest despite optimized medical therapy [14].

### 2.2. Immunohistochemistry in the Heart Tissue

Heart tissue samples from anterior or basolateral left ventricular walls were obtained from explanted hearts after transplantation and were snap frozen within less than 4 h. Frozen samples were then oriented, and 6 µ-thick cryostat sections were cut and immunostained with a panel of primary antibodies applied in TRIS-buffered saline and with the use of appropriate secondary antibodies and fast-red substrate. The following panel of primary antibodies was used: goat anti angiogenin, Santa Cruz, sc-1408 (1:150); goat anti Ang-1, Santa Cruz, sc-6319 (1:200); goat anti Ang-2, Santa Cruz, sc-7016 (1:200); rabbit anti Tie-2, Santa Cruz, sc-9026 (1:100); goat anti procollagen-I, Santa Cruz, sc-8782 (1:50). Human tonsil or nasal polyps were used as a positive control. For the negative control slides, normal goat or rabbit nonspecific immunoglobulins (Santa Cruz Biotechnology) were used. 

### 2.3. Scoring System for Immunohistochemistry

Light microscopic analysis was performed at a magnification of 630×. The immunostaining was scored in each cell compartment by KK and ADS and the average score was reported (range: − = absence of immunostaining, + = 1%–33% of immunostained cells; ++ = 34%–66% of immunostained cells; +++ = 67%–100% of immunostained cells) for the heart tissue.

### 2.4. Statistical Analysis

Data were expressed as mean ± standard deviation for functional data and median (range) for morphologic data. Differences between groups were analyzed using analysis of variance (ANOVA) for functional data. The ANOVA test was followed by the unpaired *t*-test for comparison between groups. The Mann–Whitney U test was applied for comparison between groups of morphologic data. Probability values of *p* < 0.05 were considered significant. Data analysis was performed using the Stat View SE Graphics program (Abacus Concepts Inc., Berkeley, CA, USA).

## 3. Results

### Immunohistochemistry in the Heart Tissue

Immunopositivity (I-pos) for angiogenin was frequently visible in the cardiomyocyte perinuclear space, in the subendothelial layer of endocardium and occasionally in endothelial cells and inflammatory cells infiltrating the heart tissue. I-pos for Ang-1 was mainly observed in the cardiomyocytes, and only occasionally in endothelial cells and infiltrating inflammatory cells. I-pos for Tie-2 was widely expressed in endocardial endothelial cells, cardiomyocytes, and occasionally in the inflammatory cells. I-pos for Ang-2 showed a different distribution, being mainly observed in the endothelial cells and occasionally in inflammatory cells infiltrating the heart tissue (Table 1). Though less represented in the heart tissue in comparison to other angiogenic molecules, I-pos for Ang-2 was significantly higher in ICM than NIDCM samples (Mann–Whitney: *p* = 0.03). Scored I-pos for procollagen I was similar in ICM and NIDCM samples (Table 2 and Figure 1A,B).

## 4. Discussion

In our samples of heart tissue from CHF patients, we found high levels of immunopositivity for angiogenin, Ang-1, and Tie-2 in both ICM and NIDCM patients. Previous studies reported increased pro-angiogenic biomarkers in chronic HF patients [15,16,17]. However, these reports are based on serum evaluation of angiogenic biomarkers whereas we report the presence of angiogenic pattern in human cardiac CHF tissue. In the ICM group, we found a significantly higher expression of Ang-2 compared to the NIDCM group. Different studies have reported an important role of Ang-2 in predicting negative outcomes in ischemic heart disease patients, and a study performed on adults with congenital heart disease confirmed this result well [15,16,17,18]. In our study, Ang-2 was more pronounced in heart tissue of ICM patients suggesting a different pattern of angiogenic stimulation, or at least a different pattern of altered endothelial integrity. Based on the immunohistological analysis we found a greater distribution of Ang-1 and angiogenin in cardiomyocytes, whereas Ang-2 expression was higher in endothelial cells.

Depletion of Ang-1 in cardiomyocytes contributes to a defective formation of coronary vessels during embryonic development [19]. Furthermore, overexpression of Ang-1 has shown a protective effect in cardiomyocytes against doxorubicin induced hypoxia [20]. These data suggest a protective effect of Ang-1 in cardiomyocytes. Indeed, we found a greater distribution of Ang-1 in cardiomyocytes compared to endothelial cells. In line with experimental data reporting that Ang-2 is stored in endothelial cells [21], in our series of CHF myocardial samples, we found Ang-2 in the endothelial cells and occasionally in inflammatory cells infiltrating the heart tissue. The upregulation of Ang-2 observed in the myocardial tissue of ICM patients may suggest both increased inflammatory activation and a more peculiar attempt at cardiac revascularization in this subgroup of patients with CHF. Unfortunately, the low levels of immunopositivity for Ang-2 in the heart tissue can only allow us to speculate regarding the role of this molecule in cardiac remodeling. Furthermore, the similar immunopositivity score for procollagen I in ICM and NIDCM, irrespective of the initial cause of cardiomyopathy, may suggest that despite the different intermediate molecular mechanisms, the myocardial fibrotic process in advanced HF is comparable. Our results suggest that in cardiac remodeling of ischemic and nonischemic end stage heart failure differences in angiogenic protein expression are present. Further studies on larger populations may shed light on the complex and intriguing processes leading from left ventricular systolic dysfunction to the development of heart failure.

### Study Limitation 

Our data are based on a relatively small group of patients and we did not include heart samples from healthy controls. Furthermore, we did not evaluate serum biomarkers. 

## 5. Conclusions

In our small series of CHF heart tissue samples, the cell distribution of pro-angiogenic molecules is different, Ang-1 and angiogenin being higher in cardiomyocytes, and Ang-2 higher in endothelial cells. Moreover, Ang-2 expression is more pronounced in heart tissue samples of ICM than NIDCM, suggesting a different pattern of angiogenic stimulation or, at least, a different pattern of altered endothelial integrity. These data may contribute to a better understanding of the angiogenesis signaling pathways in CHF. Further studies on a wider scale are needed to investigate in greater depth different patterns in the biochemical processes leading from ICM and NIDCM to the complex heart failure syndrome.

## Figures and Tables

**Figure 1 medicina-55-00766-f001:**
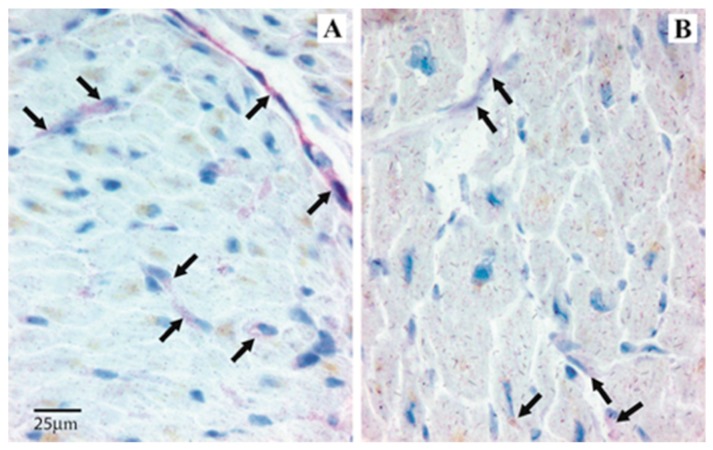
Angipoietin-2 expression in heart tissue samples from ICM and NIDCM. Immunopositivity is mainly observed in the endothelial cells (red color). Arrows indicate immunopositivity in a representative patient with ICM (**A**) and in a representative patient with NIDCM (**B**). Immunopositivity is significantly higher in the endothelial cells from ICM patients.

**Table 1 medicina-55-00766-t001:** Immunopositivity scored distribution of angiogenic molecules in heart tissue samples coming from advanced chronic heart failure (CHF) patients.

Cell Type	Angiogenin	Ang-1	Ang-2	Tie-2	PICP
Cardiomyocyte perinuclear space	+++	+++	−	+++	+++
Endothelial Cells	+−−	+−−	+++	+++	−−−
Infiltrating Inflammatory Cells	+−−	+−−	+−−	+−−	−−−

Ang: Angiopoietin, PICP, Procollagen-I. Semiquantitative evaluation for identification of sites of immunopositivity.

**Table 2 medicina-55-00766-t002:** Immunohistochemical analysis of angiogenic proteins in the heart tissue obtained at transplantation from Ischemic Cardiomyopathy (ICM) and Non-Ischemic Dilated Cardiomyopathy (NIDCM) CHF patients.

Subjects (*n*)	Angiogenin	Ang-1	Ang-2	Tie-2	PICP
ICM (9)	1.25 (0.75–1.75)	2.625 (1.25–3)	0.625 (0.25–1.25) *	1.375 (1.125–1.75)	2.25 (1.5–2.5)
NIDCM (7)	0.5 (0.375–2)	2.5 (0.25–2.75)	0.25 (0–0.75)	1.75 (0–2.125)	2.25 (1.5–2.5)

Heart tissue immunopositivity for angiogenic proteins was scored from 0 = absence of immunopositivity to 3 = extensive immunopositivity involving all endothelial cells and cardiomyocytes). Ang: Angiopoietin; ICM: Ischemic Cardiomyopathy; NIDCM: Non-Ischemic Dilated Cardiomyopathy; PICP, Procollagen-I. Data expressed as median (range); * *p* < 0.05.

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
