# Peer review of "Indexes of Angiogenic Activation in Myocardial Samples of Patients with Advanced Chronic Heart Failure"

_medicina, 2019, doi:10.3390/medicina55120766_

Round 1

Reviewer 1 Report

This is an interesting study. However the methodology and result reporting is a little confusing. Also the clinical significance of this work is uncertain, as the authors donot report an association with disease severity at time of transplant or with any serum biomarkers. Additionally the mechanism of development of the two varieties of HF is different and hence the difference in levels of angiogenesis factors may be expected.

Technical soundness of the work Table 1 is difficult to understand – are the scores the mean or median of the patients? Also, this may be better shown separately for the ischemic patients and the cardiomyopathy patients, as the degree of expression of the proteins was different in the two cohorts. Table 2 is confusing since the scoring is in ‘+’ in Table 1 and then becomes numeric in table 2. Also the exact range is not defined – instead of just mentioning range from 0 to 3, authors may include what each number meant so one can understand how the calculations were done The distribution of angiogenin and angiopoetin 1 seems similar in the table but the description in the results para makes it sound like it is different. Also the results should mention the proteins in the order that they are mentioned in the table Table 2 - may consider including p-values Line 104 is better included in discussion Figure 1 – there is no description of what one is seeing – consider including a detailed legend to describe Line 130-133 – grammar edits Line 115 – Figure 1 is not describing procollagen Clinical relevance is not described well Line 160-163 – the limitations of ‘being short communication data’ may be omitted Rigor of analysis – methodology does not mention who reviewed the slides, or if there were any differences between the patients of the two cohorts in terms of demographics, severity of HF, listing status, and duration of HF

Clear use of English language -  Line 60-64 - edits in grammar Line 81-83 – split into 2 sentences

Author Response

Reviewer # 1

This is an interesting study. However, the methodology and result reporting is a little confusing. Also the clinical significance of this work is uncertain, as the authors do not report an association with disease severity at time of transplant or with any serum biomarkers. Additionally the mechanism of development of the two varieties of HF is different and hence the difference in levels of angiogenesis factors may be expected.

We want to thank the reviewer for his/her comments, criticism and suggestion. We revised the manuscript according to reviewers suggestions and we hope that we satisfied his/her requests.

All patients suffered from end stage chronic heart failure (CHF), due to severe left ventricular systolic dysfunction, and all samples have been obtained from hearts explanted at the time of heart transplantation. As a consequence, it is impossible to further stratify the severity of the disease at this time.

In 2011 we studied 87 CHF patients, and we didn’t find any significant difference in serum levels of angiopoietin I, angiopoietin II, angiogenin, Tie-2, VEGF, and even NT-pro-BNP between ischemic and non-ischemic CHF (1). Therefore, the finding of significantly different expression of angiopoietin II in heart samples of ischemic vs non-ischemic cardiomyopathy, (with no difference in terms of expression of angiopoietin I, angiogenin and Tie-2), is least intriguing. Angiopoietin II is a complex molecule that subtends multiple processes such as neo angiogenesis and inflammation, and deserves potent prognostic significance similar to NT-pro-BNP in CHF (2). These observations led us to report the findings of the present work, as a further contribution to the comprehension of pathophysiology of CHF, and hoping it might be a trigger to further studies.

References:

Eleuteri, E., Di Stefano, A., Tarro Genta, F., et al. Stepwise increase of angiopoietin-2 serum levels is related to haemodynamic and functional impairment in stable chronic heart failure. Eur J Cardiovasc Prev Rehabil. 2011 Aug;18(4):607-14. Eleuteri et al 2016, IJC

Technical soundness of the work Table 1 is difficult to understand – are the scores the mean or median of the patients?

Also, this may be better shown separately for the ischemic patients and the cardiomyopathy patients, as the degree of expression of the proteins was different in the two cohorts.

Table 2 is confusing since the scoring is in ‘+’ in Table 1 and then becomes numeric in table 2. Also the exact range is not defined – instead of just mentioning range from 0 to 3, authors may include what each number meant so one can understand how the calculations were done.

The distribution of angiogenin and angiopoetin 1 seems similar in the table but the description in the results para makes it sound like it is different. Also the results should mention the proteins in the order that they are mentioned in the table Table 2 - may consider including p-values

Table 1 has the scope to identify the tissutal structures expressing the molecules studied. For this reason we used a semiquantitative method for identification of the immunopositivity distribution. This is now included in the Table 1 legend. In table 2 we used a quantitative approach for quantification of the molecules expressed in the two subgroups of patients. We expressed the data as median (range), as stated in the discussion section ad asterisk (angiopoietin-2, ICM) identifies statistical differences between groups. This is also stated in the Table 2 legend in the revised version of the manuscript.

Line 104 is better included in discussion

Done.

Figure 1 – there is no description of what one is seeing – consider including a detailed legend to describe

Done. We added a legend for figure 1. Please check page 5, lines 146-149.

Line 130-133 – grammar edits

Corrected. Please check lines 149-150 in the revised manuscript.

Line 115 – Figure 1 is not describing procollagen

Corrected.

 Clinical relevance is not described well

 We agree with the reviewer; to better address this comment, we have added an ad hoc comment in the Discussion

Line 160-163 – the limitations of ‘being short communication data’ may be omitted

We deleted this statement from study limitations.

Rigor of analysis methodology does not mention who reviewed the slides, or if there were any differences between the patients of the two cohorts in terms of demographics, severity of HF, listing status, and duration of HF.

We thank the reviewer for this suggestion. All patients included in our study were characterized by end-stage heart failure, maximized medical and resynchronization therapy and frequent hospitalizations for HF and no differences regarding these characteristics were present.  Unfortunately, we do not have other information about the exact duration of HF and since they were listed in Heart Transplantation Registry. Following the reviewer suggestions we added in the methods section the study population, please check lines: 77-87. The reference regarding European Society of Cardiology Guideline s for Chronic Heart Failure treatment were also added. Ref. number: 14. Slides were reviewed by the co-authors of this manuscript.

Clear use of English language - Line 60-64 - edits in grammar Line 81-83 – split into 2 sentences

We revised the English language of all manuscript and made the necessary corrections. Please check the indicated modifications in the revised version of manuscript in line 60-65; and sentences in line 81-83 were modified as follows: ''Nine male patients affected by ICM and seven NIDCM were included in our study. The mean age for ICM and NIDCM groups was respectively: 60.8±1.9 and 59.6±2.4 years. ICM patients were all males and NIDCM patients included four males''.

Reviewer 2 Report

Medicina-621669 Indexes of Angiogenic Activation in Myocardial Samples of Patients with Advanced Chronic Heart Failure

The authors investigated and presented in this short communication paper that the expression of angiogenesis markers are different between ischemic heart disease and idiopathic cardiomyopathy, suggesting a different pattern of angiogenic stimulation. Also, the distributions of these markers were different between two different etiologies. There are several concerns to be clarified.

There are so many typo and grammatical errors that should be improved.

The abbreviation of “ICM” for idiopathic cardiomyopathy might not be so appropriate, because ICM is in general used for ischemic cardiomyopathy. Also, if the patients assigned to IHD group had reduced cardiac function, they may better be called “ischemic cardiomyopathy” group.

In tables, please try not to use abbreviations including CPS, FC, and IIC. Also in Figure 1, please explain the meaning of the arrows.

In Table 2, please define the meaning of “*”. Probably this means p <0.05 with statistical significance. 

Author Response

Reviewer # 2

The authors investigated and presented in this short communication paper that the expression of angiogenesis markers are different between ischemic heart disease and idiopathic cardiomyopathy, suggesting a different pattern of angiogenic stimulation. Also, the distributions of these markers were different between two different etiologies. There are several concerns to be clarified.

We thank the reviewer for his/her useful and constructive comments and suggestion.

There are so many typo and grammatical errors that should be improved.

We apologies for the typing and grammatical errors. We revised the manuscript carefully and made necessary corrections through all the text.

The abbreviation of “ICM” for idiopathic cardiomyopathy might not be so appropriate, because ICM is in general used for ischemic cardiomyopathy. Also, if the patients assigned to IHD group had reduced cardiac function, they may better be called “ischemic cardiomyopathy” group.

 We agree with the reviewer, and for making more clear the terms and abbreviations in the revised version of manuscript we assigned ICM for ischemic cardiomyopathy and NIDCM (Non Ischemic Dilated Cardiomyopathy) instead of idiopathic cardiomyopathy.

In tables, please try not to use abbreviations including CPS, FC, and IIC. Also in Figure 1, please explain the meaning of the arrows.

Ok Done. You can find the corrections on page 4 and 5. In table 1 we replaced the abbreviations and we added a legend for figure 1.

In Table 2, please define the meaning of “*”. Probably this means p <0.05 with statistical significance. 

 Done.

Round 2

Reviewer 1 Report

The authors have addressed a majority of the comments and suggestions.

As a future direction of the study, the authors should consider including more retrospective clinical data and comparing the expression of these molecules between patients with more symptomatic or more hemodynamically significant heart failure, and potentially include data like time since diagnosis of heart failure. additionally, future studies can look at heart tissues of patients who receive VAD support to compare, as well as potentially compare these molecular expressions in the same patients at time of VAD and at time of subsequent transplant to show the effect of unloading the ventricle on the fibrosis and angiogenesis. These are clinically relevant directions that may help illustrate the importance of this work more.

Line 58 – grammar check please

Line 61 – the word ‘anyway’ is not necessary

Line 65 – there is strike-out or typo

Line 75 – how many patients in each group were already hospitalized for HF at time of undergoing transplant?

Line 92 – please mention who reviewed the slides, how many reviewers looked at each slide, and how the final score was calculated (for eg. Average of the score by 2 reviewers?)

Line 126 – spelling mistake

Line 134 – please use alternative language – perhaps the last line of the para may be placed here instead of this non-specific sentence. Also the following sentence comparing cases to controls in prior studies is unnecessary as this study only has patients in HF

Line 142 – this statement is contradicting the results

Line 145 – the overall distribution of the various molecules in the respective parts of the tissue may be stated either in the beginning of this para or the beginning of the next para – right now it is just right after a statement about the differences between ICM and NICM

Line 146 - …”has BEEN demonstrated…”

Line 149 – the prognostic potential of Ang-1 and the comparison with the studies showing protective effect is not relevant as the current study only had all patients who got transplanted and no difference in outcomes. The first 3 sentences of this para can probably be omitted and perhaps replaced with the last lines of the previous para which describe the overall tissue distribution of the proteins

Line 156 – typo

Line 161 – the part of the sentence about “..similar biochemical process subtending heart failure…” is not necessary – the authors may just state that there were differences in protein expression which may benefit from further study – the current version is verbose without adding much to the paper

Line 162 – ‘expected’ would not be the right word – further studies may shed light on this topic

Line 168 – what is the meaning of “deepening of angiogenesis signaling”

Line 174 – grammar check – ‘might give a contribution’ may be replaced by ‘adds to our knowledge of…’

Line 175 – these sentences in the conclusions are similar to line 161-165 in the discussion. Perhaps they may be omitted from the discussion and kept only in the conclusion section.

Author Response

Dear

Prof. Dr. Edgaras Stankevičius Editor In Chief

Medicina

Point-by-point authors’ response to the Reviewers’ and Editor’s comments

We would like to thank both the Reviewer for the remarks and suggestions, which helped us to improve further the quality of our manuscript, and gave us important suggestions for further studies. A native English speaker revised the English language of our manuscript.

The authors have addressed a majority of the comments and suggestions.

As a future direction of the study, the authors should consider including more retrospective clinical data and comparing the expression of these molecules between patients with more symptomatic or more hemodynamically significant heart failure, and potentially include data like time since diagnosis of heart failure. additionally, future studies can look at heart tissues of patients who receive VAD support to compare, as well as potentially compare these molecular expressions in the same patients at time of VAD and at time of subsequent transplant to show the effect of unloading the ventricle on the fibrosis and angiogenesis. These are clinically relevant directions that may help illustrate the importance of this work more.

We Thank the Reviewer for His/Her relevant suggestion. We will consider the application of our data regarding clinical significance. Would be of great interest also for a future study the role of continues flow LVAD on cardiac remodeling.

The English language of our text was revised by a native English speaker as well.

Line 58 – grammar check please

Done

Line 61 – the word ‘anyway’ is not necessary

OK.

Line 65 – there is strike-out or typo

Line 75 – how many patients in each group were already hospitalized for HF at time of undergoing transplant?

Patients undergoing transplant were recovered as planned  hospitalization, none of them was urgent transplantation.

Line 92 – please mention who reviewed the slides, how many reviewers looked at each slide, and how the final score was calculated (for eg. Average of the score by 2 reviewers?)

The immunostaining was scored in each cell compartment by KK and ADS and the average score was reported (range: − = absence of immunostaining, + = 1–33% of immunostained cells; ++ = 34–66% of immunostained cells; +++ = 67–100% of immunostained cells) for the heart tissue.

This text is reported now in line 93-94.

Line 126 – spelling mistake

Corrected.

Line 134 – please use alternative language – perhaps the last line of the para may be placed here instead of this non-specific sentence. Also the following sentence comparing cases to controls in prior studies is unnecessary as this study only has patients in HF

We deleted compared to controls.

Line 142 – this statement is contradicting the results

Apologies! We replaced less with more.

Line 145 – the overall distribution of the various molecules in the respective parts of the tissue may be stated either in the beginning of this para or the beginning of the next para – right now it is just right after a statement about the differences between ICM and NICM

Thank You. We would prefer only language modifications.

Line 146 - …”has BEEN demonstrated…”

Ok

Line 149 – the prognostic potential of Ang-1 and the comparison with the studies showing protective effect is not relevant as the current study only had all patients who got transplanted and no difference in outcomes. The first 3 sentences of this para can probably be omitted and perhaps replaced with the last lines of the previous para which describe the overall tissue distribution of the proteins.

Thank You, but in this case  we would prefer to maintain the text as it i, with the necessary englesh language corrections.

Line 156 – typo

Corrected.

Line 161 – the part of the sentence about “..similar biochemical process subtending heart failure…” is not necessary – the authors may just state that there were differences in protein expression which may benefit from further study – the current version is verbose without adding much to the paper

Thank You! We modified the text and added the following statement: Our results suggest that in cardiac remodeling of ischemic and non-ischemic end stage heart failure differences in angiogenic protein expression are present.

Line 162 – ‘expected’ would not be the right word – further studies may shed light on this topic

Ok

Line 168 – what is the meaning of “deepening of angiogenesis signaling”

This part was canceled.

Line 174 – grammar check – ‘might give a contribution’ may be replaced by ‘adds to our knowledge of…’

Ok.

Line 175 – these sentences in the conclusions are similar to line 161-165 in the discussion. Perhaps they may be omitted from the discussion and kept only in the conclusion section.

Thank You, but we would prefer to maintain the text as it is

Reviewer 2 Report

There are no further comments to be addressed. 

Author Response

Thank You!